# Ultrafast small-scale soft electromagnetic robots

Guoyong Mao [1,3] ✉, David Schiller[1,2,3], Doris Danninger [1,2], Bekele Hailegnaw[1,2], Florian Hartmann[1,2], Thomas Stockinger[1,2], Michael Drack[1,2], Nikita Arnold [1,2] & Martin Kaltenbrunner [1,2] ✉

High-speed locomotion is an essential survival strategy for animals, allowing populating harsh and unpredictable environments. Bio-inspired soft robots equally benefit from versatile and ultrafast motion but require appropriate driving mechanisms and device designs. Here, we present a class of small-scale soft electromagnetic robots made of curved elastomeric bilayers, driven by Lorentz forces acting on embedded printed liquid metal channels carrying alternating currents with driving voltages of several volts in a static magnetic field. Their dynamic resonant performance is investigated experimentally and theoretically. These robust and versatile robots can walk, run, swim, jump, steer and transport cargo. Their tethered versions reach ultra-high running speeds of 70 BL/s (body lengths per second) on 3D-corrugated substrates and 35 BL/s on arbitrary planar substrates while their maximum swimming speed is 4.8 BL/s in water. Moreover, prototype untethered versions run and swim at a maximum speed of 2.1 BL/s and 1.8 BL/s, respectively.

Natural organisms, such as cheetahs, rabbits, or cockroaches, use high-speed locomotion as one of their main survival strategies to hunt for food or flee from predators. The relative speed in terms of body lengths (BL) per second quantifies the velocity of different organisms across a large spectrum of body sizes, and can be as high as 323 BL/s for the mite *Paratarsotomus macropalpis*[1]. The technology achieves high-speed locomotion mainly through large-scale machines (BL > 100 mm) and high-power engines (such as combustion- or electric motors), resulting in Formula One cars (50 BL/s) or quadrupedal robots[2] running at 9.1 BL/s. However, the design of high-speed small-scale robots (1 mm < BL ≤ 100 mm) is challenging because of the difficulties in the miniaturization of traditional high-performance motors and transmission systems. Simple structures made of smart materials provide alternative possibilities to build miniaturized robots. Lead zirconate titanate (PZT)[3] and shape memory alloys (SMA)[4] are two representative rigid smart materials implemented in millimeter-sized robots, but feature either too small actuation strokes or low frequencies to allow high-speed locomotion. Emerging robotics and human-robot interaction in addition require soft, safe, fast and robust designs capable of

operation in harsh, dynamic environments. An extreme example is a human stomach, undergoing mechanical compression during digestion and containing acidic fluids. Preventing or treating gastrointestinal tract-related diseases promotes the development of soft mini-robots for drug delivery or non-invasive surgery[5].

To tackle these problems, soft smart materials for robotics, such as thermo-responsive polymer fibers[6], pH-responsive polymer gels[7], light-responsive liquid crystal polymers[8] and electric/magnetic field-responsive materials[9–13] have emerged. However, thermo-responsive polymer fibers and pH-responsive polymer gels rely on the slow diffusion of ions or heat and are thus not fast enough for high-speed locomotion in robots. Light-responsive liquid crystal polymers[8] can be actuated at frequencies exceeding 10 Hz, but the need for modulated illumination and transparent environments restricts their application possibilities[14]. Electric/magnetic field-responsive elastomers, such as dielectric elastomers (DE) and soft magnetic elastomers (SME), typically feature fast response times, with vibrations in the kHz range[9–12]. Drawbacks of DEs are their high actuation voltages (in the kV range), posing potential safety issues and impeding miniaturization. SME

[1]Soft Materials Lab, Linz Institute of Technology, Johannes Kepler University, Altenberger Str. 69, 4040 Linz, Austria. [2]Division of Soft Matter Physics, Institute for Experimental Physics, Johannes Kepler University, Altenberger Str. 69, 4040 Linz, Austria. [3]These authors contributed equally: Guoyong Mao, David Schiller. ✉e-mail: guoyong.mao@jku.at; martin.kaltenbrunner@jku.at

robots are safe, have fast response, and are easy to miniaturize, but have difficulties with multiple-module or swarm robot designs, as they require global, dynamically tunable magnetic fields[14–16]. Soft electromagnetic actuators (SEMA) comprising liquid metal (LM) coils embedded in elastomeric substrates have better local controllability and feature high performance in a strong static magnetic field, as exists e.g., in a magnetic resonance imaging (MRI) machine[9]. Advances in LM 3D printing allow the miniaturization of SEMAs to at least a millimeter-scale, opening up routes towards high-speed locomotion in micrometer- to centimeter-sized soft robots[17].

Here we develop a series of ultrafast, robust, and versatile small-scale soft electromagnetic robots (SEMRs) capable of walking, running, jumping, swimming, steering and even transporting and releasing cargo. This is achieved through advancements in fabrication, robot design, and modeling, which collectively boost the robot performance and even permit untethered operation when equipped with a miniaturized self-powered controller. The fabrication (Fig. 1a) utilizes printing of LM coils on elastomeric substrates, which allows simultaneous selective control of different sections of the robot body, enabling steering and transporting cargo. An elastomeric bilayer with strain mismatch results in a curved robot body capable of walking

when subjected to a time-varying current in a static magnetic field and proper feet design. We introduce two types of SEMR feet: sawtooth-shaped feet for asymmetric 3D printed substrates, and L-shaped feet for planar substrates. Both are shown in Fig. 1a with SEMR TST (tethered, sawtooth-shaped feet) and SEMR TL (tethered, L-shaped feet). The locomotion of SEMRs becomes ultrafast near the mechanical resonance frequency, as demonstrated in experiments and supported by analytic and numerical modeling. We demonstrate SEMRs running at an ultrafast relative speed of 70 BL/s, about 17.5 times faster than previous soft-bodied robots, faster than centimeter-scale electromagnetic robots and most animals (Fig. 1b). Furthermore, the same SEMR is also able to swim at a high relative speed of 4.8 BL/s, which compares favorably to other aquatic robots and animals (Supplementary Fig. 1, Supplementary Table 2).

## Results and discussion
### Validation of fabrication

The versatile design of the LM channels within SEMRs is provided by a 3D direct ink writing (DIW) printer[18] (Supplementary Fig. 2) capable of producing LM wires down to 100 μm resolution (Supplementary Fig. 3). To connect the LM channels to external power, we insert

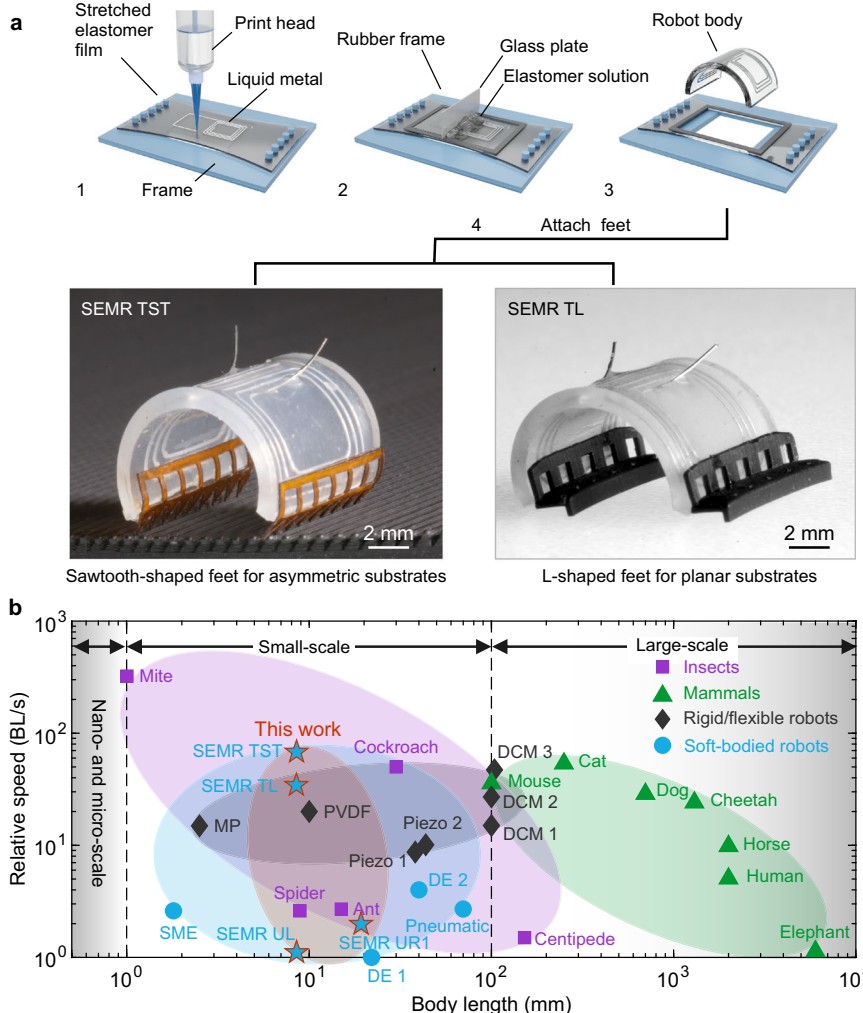

**Fig. 1 | Fabrication and performance of SEMRs. a** Schematic of the fabrication process. LM coils are printed on a prestretched elastomeric film. Then, the elastomer precursor solution is applied on top of the LM coils using bar coating, resulting in a bilayer structure from which the robot body is cut out. Lastly, sawtooth-shaped or L-shaped feet are attached. For actuation, the robots are connected to the external power with electrodes. **b** Maximum running speeds of representative mammals, arthropods, soft robots, and robots versus body length.

Shaded areas encompass the ranges for different categories, as indicated by the symbols in the legend, and for our SEMRs, which are labeled with the stars. The maximum relative speed of our SEMRs is 70 BL/s, almost 17.5 times larger than for the previous soft-bodied robots, faster than centimeter-scale electromagnetic robots and most fast animals. Two stars with the higher speed correspond to tethered SEMRs, and two slower ones to untethered robots. Details can be found in Supplementary Table 1.

electrodes into the SEMA/SEMR body and seal them with elastomer (Supplementary Fig. 4). The robustness and functionality of our fabrication scheme were investigated in bending tests with two small-scale SEMAs with a size of 9 mm × 9 mm × 0.8 mm (Supplementary Fig. 5). Here, SEMA 1 weighs 96 mg, while SEMA 2 is even lighter due to a cutout in the center. Both actuators are driven by a custom pulse-width modulation (PWM) controller (Supplementary Fig. 6) and are placed on top of a permanent plate magnet (magnetic field around 0.3 T at its surface, Supplementary Fig. 7a). While the horizontal displacement of its tip is 4.5 mm for SEMA 1 when driven by a constant current of 1 A, the lighter design of SEMA 2 allows for an increased displacement of up to 6.4 mm due to a decrease in bending stiffness (Supplementary Fig. 7b–d). Dynamic tests with the SEMAs driven by a square-wave current (Supplementary Fig. 7e, Supplementary Movie 1) yield a high horizontal span of more than 6.3 mm with a relatively small current (0.1 A, 8 Hz) for SEMA 2 with the cutout geometry (Supplementary Fig. 7f). We characterize SEMA temperature increase when operated in the air (Supplementary Fig. 8a), as this is crucial for high driving currents. We find a temperature rise of 1.3 °C, 10.9 °C, and 27.5 °C for currents of 0.1 A, 0.3 A, and 0.5 A in long-term tests (>1000 s). The actuators remain fully functional and this Joule heating can be further decreased with a better coil design, such as increasing the number of coil turns, as discussed in the Supplementary Text.

## Design and fabrication of the curved SEMR body

The SEMR design is based on the principle of a 2-dimensional SEMA, but with essential modifications that endow them with a high-speed locomotion mechanism. The flat SEMAs shown in Supplementary Fig. 9a are only capable of small in-plane deformations for geometries where the LM coil is oriented perpendicular to the direction of the magnetic field. Under these conditions, the robot has difficulties deforming, let alone walking. Considering that many animals and most soft robots utilize the expansion/contraction of their curved body[8,12,19,20] for fast locomotion, we hypothesized that curved elastomeric bilayer films (Supplementary Fig. 9b) with embedded LM channels will enable high-speed soft electromagnetic actuators. For this, mismatched strains in the bilayer film are essential. Typical strategies include the utilization of pH-, thermal-, or humidity- responsive materials. Given the inherent Joule heating of SEMAs and that their typical working environment is ambient air, we concluded that those material classes may however be suboptimal to achieve curvature in SEMAs. Instead, we apply mechanical prestretch to one of the layers of the bilayer film to induce strain mismatch. This method finds application in stretchable electronics and can be scaled down to the micrometer scale[21,22]. In practice, we fabricated bilayers by bonding a prestretched layer (top) to an undeformed layer (bottom), such that when the bilayer film is released, it curls up (Supplementary Fig. 10a). To guide the fabrication and find out the desired thicknesses and prestretches for the bilayer films, a numerical finite element method (FEM) scheme (Fig. 2a, Supplementary Fig. 10b) and a theoretical model (Supplementary Fig. 10c, d) have been developed.

The theoretical radius of the bilayer film is approximated by $r = (t_{10} + t_{20})^3 / (6\varepsilon_{10} t_{10} t_{20})$, where $t_{10}$ and $t_{20}$ are the thicknesses of the top and bottom layer and $\varepsilon_{10}$ is the prestrain of the top layer before its

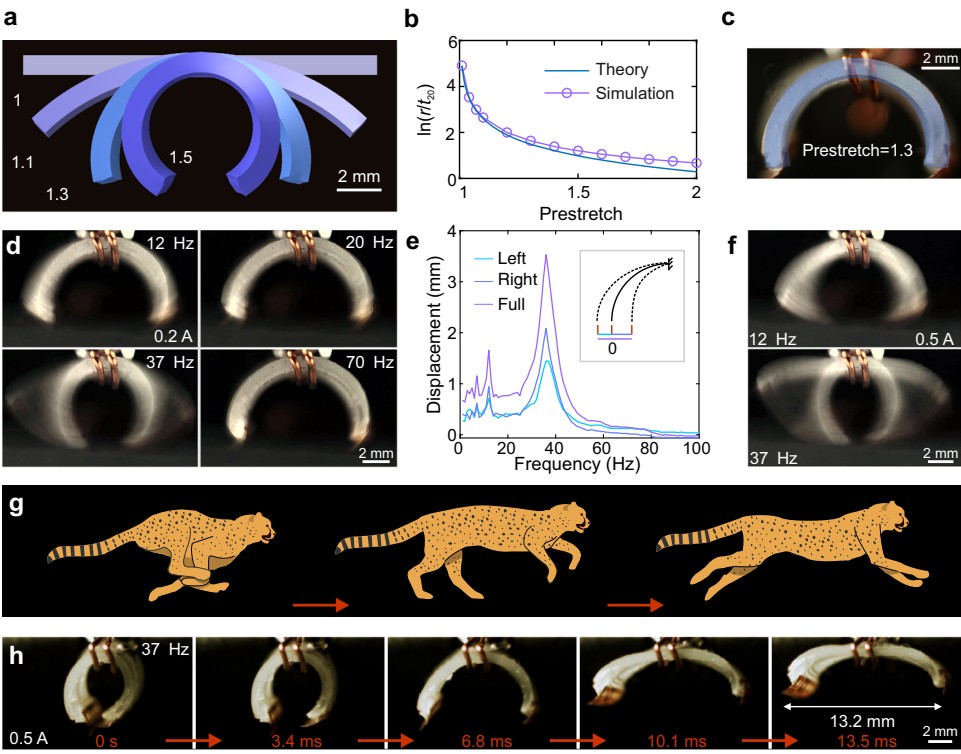

**Fig. 2 | Design and characterization of small-scale SEMRs. a** Simulated shape of the bilayer film with different prestretches: 1.0, 1.1, 1.3, and 1.5. **b** Simulated and calculated radius of the bilayer film as a function of the applied prestretch. **c** Superimposed pictures of the experimental and simulated SEMR shapes (side view) with a prestretch of 1.3. **d** Snapshots of the SEMR vibrations for 0.2 A square-wave current with different frequencies (Supplementary Movie 2). The 37 Hz and 12 Hz frames correspond to the main and the second-largest spectral maxima, which can be seen in **e**. **e** Horizontal displacement of the left foot of the robot subjected to a 0.2 A square-wave current at different frequencies (Supplementary Movie 2). The robot is mounted on the top of a magnet and clamped in the middle with copper wires. The inset illustrates three displacements (Left, Right and Full). They correspond to the maximum displacement from the reference "0" position (no current) to the left (extension), right (contraction) and their sum, respectively. The top curve (Full) shows the full range of the foot displacement. **f** Frames from the vibration test for 0.5 A square-wave currents at frequencies of 12 Hz and 37 Hz (Supplementary Movie 2) illustrate the range of motion away and close to the resonance frequency. **g** Illustration depicting a running cheetah. **h** Snapshots from the high-speed camera video (Supplementary Movie 2), which show stages of the robot movement driven by a square-wave current (0.5 A, 37 Hz).

bonding to the bottom layer (see Supplementary Text for details). The relation between the prestrain $\varepsilon_{10}$ and the prestretch $\lambda_{pre}$ is $\lambda_{pre} = 1 + \varepsilon_{10}$. If $t_{10} = t_{20}$, the radius simplifies to $r = 4t_{10}/(3\varepsilon_{10})$. In Supplementary Fig. 10e, we sketch curved bilayer films with prestretches varying from 1.01 to 1.7. The theoretical model agrees with the numerical simulations for small prestretches. (Fig. 2b). Additionally, we conducted experiments to compare three types of prestretches (equibiaxial, pure shear, and uniaxial) and found good agreement with numerical simulations (Supplementary Fig. 11). Equibiaxial stretch introduces curvature also in the 2nd direction, an undesired effect for these types of SEMRs, because it decreases the effective Lorentz force in the walking direction and complicates the motion, control and fabrication. From these prestretch types, the uniaxial one turned out to be the most practical, even though the other two produce larger curvature at the same prestretch. Therefore, we use uniaxial stretch to fabricate curved SEMRs with the aid of theory and simulation. A 3D printed frame is used to control the prestretch of a rectangular film (Supplementary Fig. 13a, b), while a small cuboid serves as a flat support substrate beneath the stretched film during the 3D printing process (Supplementary Fig. 13c–e). The SEMR manufacturing is finalized by attaching two robot feet (Fig. 1a). The fabricated SEMR has the same shape as calculated (Fig. 2c).

## Vibration of SEMRs and prediction of running speed

The physical picture of SEMR locomotion and dynamic performance is described within the theoretical model based on mechanical vibrations. A series of experiments were conducted to validate this model and further characterize the geometry (Supplementary Fig. 14a–c) and mechanical properties of the SEMR. Gravity is not considered, since its influence on vibrations is minor (Supplementary Fig. 14d). The deformation of the SEMR subjected to static loads is illustrated in Supplementary Fig. 15. By applying square-wave or sinusoidal currents to the SEMR suspended above the magnet, we study the frequency response of the resulting dynamic deformation which agrees well with the theoretical predictions (Supplementary Text). The largest deflection for a square-wave current occurs at a resonant frequency of 37 Hz (Fig. 2d–f, Supplementary Movie 2). This frequency is given by Eq. (21) in the Supplementary Text, and its influence on the robot movement is discussed in section 2.10 "Oscillator approximation and velocity". A larger current (0.5 A) corresponds to a larger deformation (Fig. 2f, Supplementary Movie 2) until the body of the SEMR stretches out almost flat at maximum swing. Both the experiments (Supplementary Figs. 16, 17) and the theory (Supplementary Text) indicate that the square-wave excitation is beneficial in several respects, especially at low frequencies. Compared to sinusoidal currents with the same amplitude, larger deformations are possible for a square-wave current; the Lorentz force reaches its maximum faster; significantly higher accelerations are provided, both factors pull the robot out of the grooves on structured substrates and overcome static friction, kick-starting the movement (Supplementary Text). Electronic implementation of the square-wave current is easier as well. Vibrating SEMRs exhibit dynamics similar to a running cheetah (Fig. 2g, h), which inspired the development of the ultra-fast running robot. Under idealized conditions, the theory predicts extremely high running speeds (Supplementary Text); the measured speed is smaller due to the slip of the feet on the substrate, deviations from the straight motion, anti-phase repulsion from the ground in the hovering regime and other detrimental effects.

## Design of robot feet and characterization of locomotion

It is well known that the paws play an important role in the high-speed running of cheetahs. Similarly, a proper feet design is crucial for the high running speed of the SEMR. We introduce two strategies for the feet design demonstrated in Fig. 1a, which are based on the mechanical analysis of the SEMR. Figure 3a shows the Lorentz forces act on

different parts of the liquid metal coils, the majority of which cancel. The free-body diagram (Fig. 3b, side view) also includes normal supporting forces ($F_{s1}$, $F_{s2}$) and frictional forces ($F_{f1} = fF_{s1}$, $F_{f2} = fF_{s2}$) where $f$ is the coefficient of dry friction ($0.1 < f < 0.5$ in typical cases). The mass of the robot is about $m = 180$ mg, resulting in a gravitational force of $G = mg = 1.8$ mN, assuming the acceleration due to gravity $g \approx 10$ m/s$^2$. The relevant Lorentz forces acting on the robot legs are horizontal; at a current $I = 0.5$ A and a magnetic field strength $B = 0.3$ T they are about $F_{1L} = F_{2R} = BIL = 2.7$ mN, where $L = (5 + 6 + 7)$ mm $= 18$ mm is the total length of the conducting wires (Fig. 3a, more details in the Supplementary Text, section 1.3, "Calculation of the Lorentz force"). The normal reaction and static friction forces are distributed approximately equally between both feet: $F_{f1} = fF_{s1} \approx fG/2$. For large currents, e.g., $I = 0.5$ A, $F_{f1} < F_{1L}$, $F_{f2} < F_{2R}$ and the robot cannot move properly due to foot slip[15]. Also, the symmetry of the coils and SEMR body induces only vibration around its center of mass without its horizontal displacement when subjected to an oscillatory current. For this reason, most soft robots use hooks[19,23], thin films[12], or plastic/elastomer composites[24] as feet to avoid slipping, break the symmetry of friction force, and allow for translational locomotion. Manufacturing robust hooks for our small-scale SEMR is challenging and unidirectional motion requires a certain asymmetry, which is realized in two ways. In the first approach, the robot feet are made of thin sawtooth-shaped polymer films; such robots run on the asymmetrically structured substrate (Fig. 1a), which provides unidirectional friction (Fig. 3c–e, i). In the second design, the asymmetry is solely due to the L-shaped feet of the robot itself (Fig. 3f–h, j). Thus such SEMRs run on a wide variety of planar unstructured substrates (Fig. 3k).

Figure 3c illustrates the working mechanism of the SEMR TST. The feet are oriented in the same direction on both legs to ensure unidirectional movement. Such feet, however, still slip on planar substrates, an issue that may be overcome with alternative materials or designs[25]. Slippage reduces on rough or corrugated substrates, which we mimic using the sawtooth-shaped substrates (Fig. 3c, Supplementary Fig. 18), to study the performance of our robots under controlled conditions. Much like the non-retractable claws of a cheetah, mechanical interlocking between the robot feet and such substrates results in highly asymmetric friction, enabling ultrafast locomotion. An oscillating current (sinusoidal or square-wave) causes the robot to periodically contract and expand its body (Fig. 3c, Supplementary Fig. 19). When the robot expands, the front foot moves forward, while the rear foot is fixed due to the mechanical interlock. Then the robot contracts, now with the front foot fixed, while the rear foot pulls up forward. These stages are shown schematically in Fig. 3c and can be seen in Fig. 3d or Supplementary Movie 3 from the experiments where the SEMR TST is driven by a square-wave current (0.3 A, 1 Hz). We manufactured 3D-printed substrates with different sawtooth profiles (height of sawtooth from $a = 0.6$ mm to 1 mm) and compared their performance. The best performing one ($a = 0.8$ mm, Supplementary Figs. 18 and 20a) was used in the subsequent experiments.

To achieve faster locomotion, an optimal driving current is critical. With a low-frequency square-wave current, the robot only moves for tens of milliseconds after a change in current direction and retains its shape even when the current is non-zero (Fig. 3d, Supplementary Movie 3). Thus, it is not surprising that the speed of the robot increases for a higher frequency of the driving current. Tests at different frequencies and various current amplitudes (Fig. 3i, Supplementary Fig. 20b) reveal that the maximum robot speed is reached at a resonant frequency $f_0 \approx 45$ Hz. When the frequency of the driving current $f_I$ is detuned from $f_0$, the speed of the robot decreases. The resonant frequency in Fig. 3i (45 Hz) is larger than in the vibration test (~37 Hz in Fig. 2e) because of the different boundary conditions for the free and clamped robot (in excellent agreement with our theory in

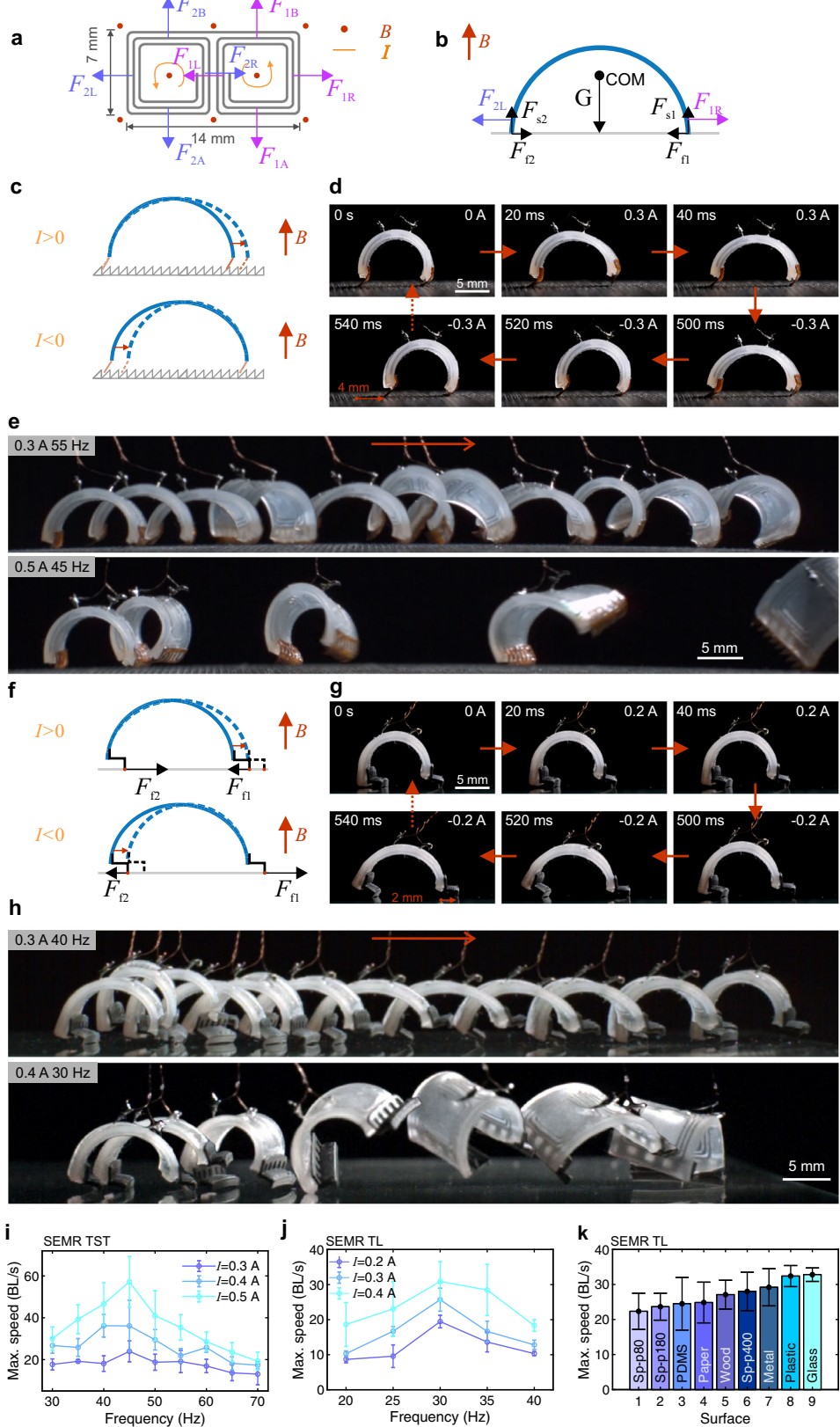

the Supplementary Text). The highest measured running speed is 630 mm/s, or 70 BL/s (Supplementary Movie 3), which is a record, 17.5 times larger than that of previous soft-bodied robots as far as we know (Fig. 1b, Supplementary Table 1). Individual frames from the running video are shown in Fig. 3e. From the curves of the displacements and velocities vs. time, we see that for lower currents, the contact and the

friction between the feet and the substrate slow the robot down (Supplementary Figs. 21a–c). However, with high currents around the resonant frequency, the robot hovers in the air most of the time or touches the ground with one foot only, which reduces the energy dissipation between the robot and the substrate and increases the running speed (Supplementary Fig. 21d). For larger magnets, the speed

**Fig. 3 | Running mechanism and performance of SEMRs. a** Lorentz forces acting on the various parts of the liquid metal coils in the external magnetic field ($B$), top view; current ($I$) is indicated by the counterclockwise arrows. The force pairs ($F_{1A}$, $F_{1B}$) and ($F_{2A}$, $F_{2B}$) are perpendicular to the bending direction and cancel. The central pair ($F_{1L}$, $F_{2R}$) also cancels in the overall balance. **b** Free-body diagram of curved SEMR with the remaining relevant loading Lorentz forces ($F_{1R}$, $F_{2L}$), side view. Gravity force ($G$) is applied to the center of mass (COM); the normal supporting forces ($F_{s1}$, $F_{s2}$) and frictional forces ($F_{f1}$, $F_{f2}$) are indicated as well. **c** Running mechanism of the SEMR TST on an asymmetrically structured substrate. **d** Key stages for the walking SEMR TST driven by a square-wave current (0.3 A, 1 Hz). **e** Snapshots of the running SEMR TSTS driven by square-wave currents (0.3 A, 55 Hz and 0.5 A, 45 Hz), as indicated. The time between the snapshots is 0.05 s. The bottom sequence corresponds to the maximum speed of 70 BL/s. **f** Running mechanism of the SEMR with the L-shaped feet. **g** Key stages for the walking SEMR TST driven by a square-wave current (0.2 A, 1 Hz). **h** Snapshots of the running SEMR TL driven by square-wave currents (0.3 A, 40 Hz and 0.4 A, 30 Hz), as indicated. The time between the snapshots is 0.05 s. The bottom sequence corresponds to the maximum speed of 35 BL/s. **i** Maximum speed of the SEMR TST driven by square-wave currents as a function of frequency at different amplitudes (0.3 A, 0.4 A and 0.5 A). **j** Maximum speed of the SEMR TL driven by square-wave currents as a function of frequency at different amplitudes (0.2 A, 0.3 A and 0.4 A). **k** Maximum running speed of SEMR TL on different substrates, including various sandpapers (Sp-p80, p180 and p400), elastomer (PDMS), paper, wood, metal, plastic and glass (Supplementary Movie 3). All error bars represent the standard deviation of four measurements. All snapshots are from different parts of Supplementary Movie 3.

should ultimately stabilize near a maximum value discussed in depth in the Supplementary Text for various mechanisms. Though the robot speed increases with current, at very high currents (above 0.6 A) the robot trips over, due to excessive folding of its body (Supplementary Fig. 22a). We suggest three solutions to this issue: (1) shortening the duration of negative current (contraction) (Supplementary Fig. 22b); (2) decreasing the amplitude of negative current (Supplementary Fig. 22c); (3) increasing the driving frequency. All these solutions work well (Supplementary Movie 4) and the speed of the robot may increase even more with further optimization.

Although we have achieved a record-high running speed using the sawtooth-shaped feet, the dependence on substrate properties limits the applicability of SEMRs. To overcome this, we developed the L-shaped feet design which is much more universal. The working mechanism of the SEMR TL employing the L-shaped feet is illustrated in Fig. 3f and Supplementary Fig. 23. The relatively large L-shaped feet are attached on the inside of the rear leg and the outside of the front leg (Supplementary Figs. 24a, b). This built-in asymmetry alternatively shifts the weight between the feet such that the normal reaction and static friction forces are distributed unequally, akin to the human walking and running cycle. For positive currents $I > 0$ during the expansion (Fig. 3f, upper panel), the front (right) foot has small friction $F_{f1} = fF_{s1} \approx 0$ and slips forward (to the right), while the rear (left) foot has large friction $F_{f2} = fF_{s2} \approx fG$ and is almost fixed. For negative currents $I < 0$ during the contraction (Fig. 3f, lower panel) the situation is reversed: the front foot has large friction $F_{f1} = fF_{s1} \approx fG$ and is almost fixed, while the rear one where $F_{f2} = fF_{s2} \approx 0$ pulls up. A detailed explanation of this behavior is given in the Supplementary Text, section 1.17, "Locomotion principle of the SEMR with the L-shaped feet". One can see these stages in the frames of Fig. 3g (taken from the Supplementary Movie 3) for the square-wave excitation at 1 Hz by a low current of 0.2 A. The resonant frequency of the SEMR TL has been measured (Supplementary Fig. 24c) to be lower than that of SEMR TST, because of the additional weight of the L-shaped feet. Six different geometries of L-shaped feet (Supplementary Fig. 24a, Supplementary Table 3) were tested and the fastest foot type E (Supplementary Fig. 24d) was selected for the subsequent experiments.

Figure 3h shows frames from Supplementary Movie 3 where SEMR TL runs on a glass plate. The upper panel (0.3 A, 40 Hz) demonstrates controlled running. After a short acceleration stage, the velocity stabilizes at a constant value of 165 mm/s, corresponding to 18.3 BL/s. The bottom sequence has a higher current and is closer to the mechanical resonance (0.4 A, 30 Hz); this leads to a much faster movement (630 mm/s, 70 BL/s), but the motion is less controlled. The frequency dependences for different currents in Fig. 3j demonstrate resonant behavior, similar to that in Fig. 3i. The resonant frequencies differ between the two designs due to differences in robot dimensions and weight (Supplementary Fig. 24c). Figure 3k lists the maximal speeds achieved by the SEMR TL on various substrates with different tribological properties under resonant conditions. The displacement vs. time curves for the SEMR TL shows stable locomotion on most

substrates, especially smooth ones like glass and metal (Supplementary Fig. 25).

## Multifunctionality: more than speed

Besides moving at high speeds, many animals developed a wide range of survival strategies including resilience to impact or falling, the ability to evade obstacles, cross the terrestrial/aquatic border at will, as well as transport prey and/or offspring. Some of these capabilities inspire robotics, where e.g., high durability increases robot survival rate in harsh environments[19,26]. We conducted durability tests in which our SEMRs are flattened by high force (1764 times its body weight) during walking (Fig. 4a, Supplementary Movie 5). The robot performance remains consistent before and after the impact. We even compressed the body of SEMR TL with a tensile test machine and found that the sealed LM can sustain compressive stresses up to 139 atm (2000 N on body). Electrical disconnection occurred at pressures exceeding 3.5 atm (50 N force onto the body), but the resistance and the body of SEMR recovered when the pressure was released (Supplementary Fig. 26).

Yet, resilience alone is not enough; overcoming obstacles is still challenging for most small-scale robots[15]. We demonstrate that our SEMR TST can jump over obstacles with a height of up to 4 mm (about 2/3 its height, Supplementary Fig. 27a, Supplementary Movie 6) by applying a pulsed current signal (100 ms negative current of −1 A, followed by 50 ms positive current of 1 A). Frame-by-frame analysis (Supplementary Fig. 27a) shows that the robot first contracts and then tilts its body (storing energy in the process). Then it expands, like a compressed spring (releasing energy), and jumps over the obstacle. Similarly, the SEMR TL can jump 3 mm upwards on the metal surface (Supplementary Fig. 27b, Supplementary Movie 6), as well as across an obstacle, or jump onto a stage and continue running (Fig. 4b, Supplementary Movie 6). Lifting the constraint of terrestrial environments for small-scale robots to include aqueous working conditions allows for manipulation of floating objects for micromanufacturing[27]. Amphibious SEMRs, due to their controllability, would enable a wide range of applications. With its light weight and relatively low average density (about $1.2 \, \text{g cm}^{-3}$), the SEMR TST floats on a water surface without further modification due to surface tension and buoyancy (Fig. 4c). When actuated with a square-wave current (0.5 A, 20 Hz) the robot swims along the water surface (Fig. 4d) at a maximum speed of 43 mm/s or 4.8 BL/s (Fig. 4e, Supplementary Movie 7), which compares favorably to other swimming robots (Supplementary Fig. 1, Supplementary Table 2). Further improvements may allow the SEMR to reach the speed of certain insects (136.4 BL/s), fishes (17.8 BL/s), or tadpoles (17.9 BL/s). Complex tasks in robotics usually involve several degrees of freedom that are often difficult to realize in small-scale soft robots due to the required control mechanisms. Since the printed LM channels of SEMRs are individually addressable and scalable, a two-module SEMR TSTS (tethered sawtooth-shaped feet, steering) containing two separate coils (Fig. 4f, Supplementary Fig. 28) is already steerable. Independent control of the coil currents allows the robot to walk

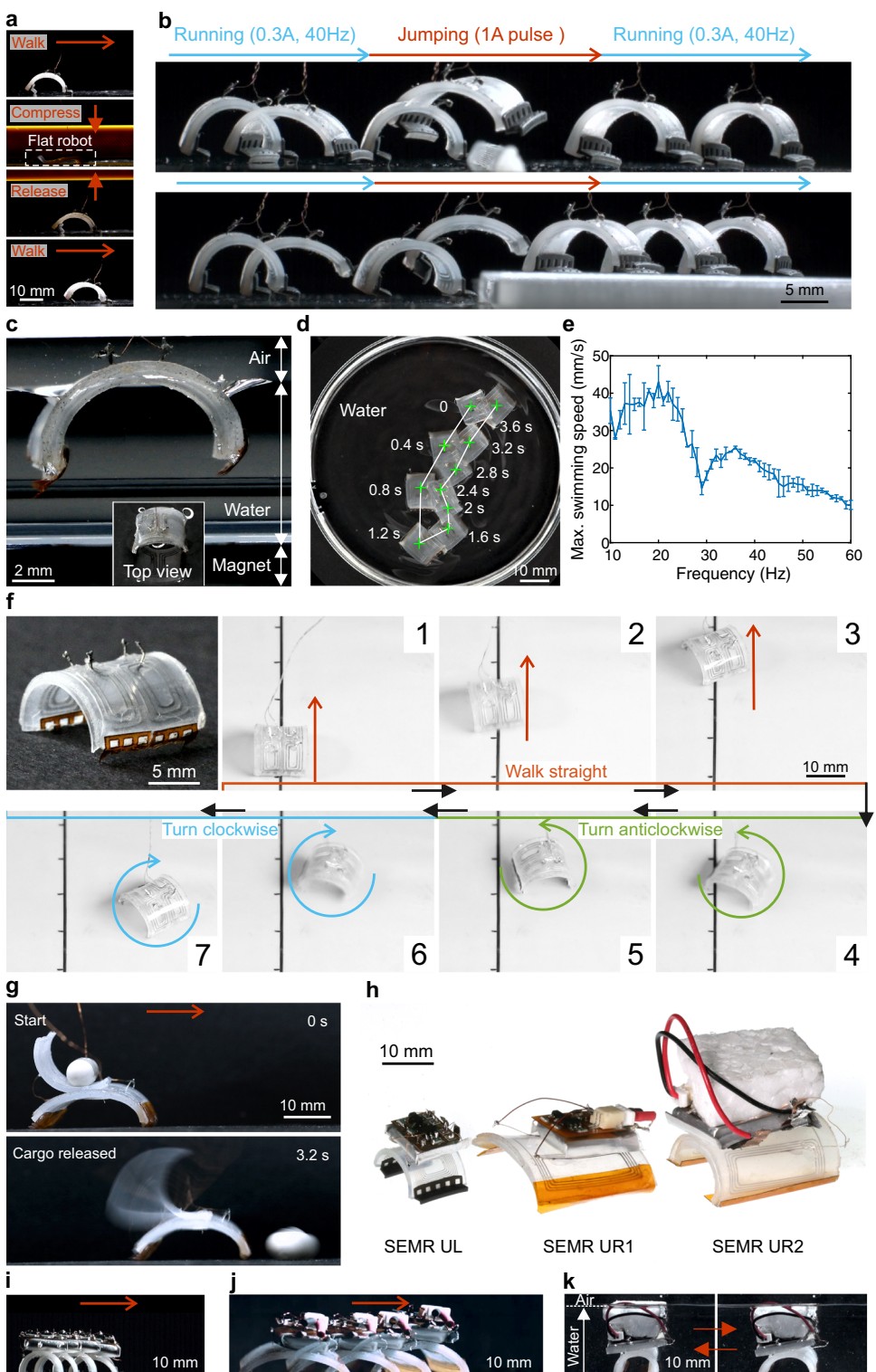

**Fig. 4 | Robust and versatile SEMRs: jumping, swimming, navigating and autonomy. a** Robustness test showing the resilience of SEMR TL upon external loading (Supplementary Movie 5). The robot resumes walking after being pressed and fully flattened twice (driving current, 0.3 A, 1 Hz). **b** The SEMR jumps across (top one) and onto (bottom one) a 2.5-mm-high object (Supplementary Movie 6). **c** SEMR TST floating on the surface of the water. Inset, top view of the robot. **d** Path of the swimming SEMR TST (Supplementary Movie 7). **e** Maximum swimming speed versus frequency for a driving current of 0.5 A. Error bars represent the standard deviation of four measurements. **f** Two-module steerable SEMR TSTS. Frames 1 to 7 show straight walking, anticlockwise, and clockwise turns using controlled currents through the modules (Supplementary Movie 8). The tick interval along the straight line is 1 cm. **g** Side view of the transport SEMR TRC, carrying cargo. It consists of a body and a release actuator for automated cargo handling (Supplementary Movie 9). **h** Photos of untethered SEMRs UL, UR1 and UR2, from left to right. **i** A sequence of snapshots of the running untethered SEMR UL (Supplementary Movie 10) with a time interval of 0.6 s. **j** A sequence of snapshots of the running untethered SEMR UR1 (Supplementary Movie 10) with a time interval of 0.27 s. **k** Snapshots of the swimming SEMR UR2 (Supplementary Movie 10) in states of contraction and expansion, left and right subpanel, respectively.

straight, turn clockwise or anticlockwise (Fig. 4f, Supplementary Movie 8) with an angular velocity of 160°/s and thus navigate freely. A further structural redesign of the two-module SEMR that includes reconfiguring the position of the second coil results in the transport SEMR TRC (tethered, rectangular feet, cargo) (Supplementary Fig. 29). Automatic relocation of objects becomes possible by actuating the two modules individually, one for running and the other one for controlled release of the cargo (Fig. 4g, Supplementary Movie 9).

## Prototype untethered SEMRs

Up to this point, the tethered SEMRs were driven by external power systems. However, self-powered/untethered robots possess larger navigational freedom and may respond more readily to surroundings or carry out general-purpose tasks. Despite the difficulties of reaching energetic and computational autonomy in small-scale systems, this step is crucial for achieving the ultimate dream of autonomous, self-propelled microrobots. Herein, we provide a straightforward prototyping solution towards untethered SEMRs by replacing the cargo manipulation actuator of a transport SEMR with a custom battery-powered printed circuit board (PCB) (Materials and Methods). Three different sizes of PCBs (Supplementary Figs. 30–32) and non-magnetic batteries (Supplementary Table 4) are used to build the controllers. In Fig. 4h, we show a group picture of the untethered SEMRs. The body length of SEMR UL (untethered, L-shaped feet) is 9 mm and around 20 mm for SEMR UR1 (untethered, rectangular feet, No. 1) and UR2 (untethered, rectangular feet, No. 2). The summary of these robots is provided in Supplementary Table 5. The SEMR UL runs on a metal surface (Fig. 4i, Supplementary Fig. 33) at a speed of 1.2 BL/s. The SEMR UR1 can run at a speed of 2.1 BL/s on a 3D printed substrate (Fig. 4j) and swim at a speed of 1.25 BL/s in water (Supplementary Fig. 34). The high internal resistance of the batteries (Supplementary Table 4) and the positive-only square-wave currents delivered by the small and medium PCBs (Supplementary Fig. 30 and Supplementary Fig. 31) limit the running performance of the SEMRs. Consequently, we developed a larger controller, including PCB (Supplementary Fig. 32) and battery (Supplementary Fig. 35a, Supplementary Table 4) capable of alternating square-wave current (Supplementary Fig. 35b). The SEMR UR2 with the large PCB and battery (Supplementary Fig. 35c–f) can swim at a maximum speed of 1.8 BL/s (Fig. 4k, Supplementary Fig. 35g, h, Supplementary Movie 10). The comparisons between untethered SEMRs and other robots in Fig. 1b, Supplementary Fig. 1, Supplementary Table 1 and Supplementary Table 2 lead to the conclusion that our SEMRs possess high speeds for both running and swimming. However, the untethered SEMRs are slower than the tethered ones, because of an increase in weight and size (Supplementary Table 5), a simplified controller design, low performance of the battery, all of which can benefit from further optimization. More details about the untethered SEMRs are in the Supplementary Text.

In summary, we have presented small-scale SEMRs with ultrahigh speed (up to 70 BL/s), featuring high robustness, multimodal locomotion and untethered operation that render them highly suitable for versatile applications in electrically-controlled intelligent systems. Furthermore, stronger magnetic fields, such as the interior of an MRI machine[28], will greatly enhance the speed, power output and efficiency of SEMRs[9]. Straightforward and scalable fabrication using 3D direct ink writing renders them highly suitable for versatile applications in electrically-controlled intelligent systems and empowers the development of future high-performance microrobots for flexible microfabrication, targeted drug delivery and non-invasive surgery, where agility is of paramount importance[5,29,30].

## Methods

### Elastomer

The elastomer for the fabrication of bilayer films is prepared by mixing Ecoflex 00-30 (Smooth-On Inc.) and Polydimethylsiloxane (PDMS) (Sylgard 184, Dow Corning Inc.) solution with a mass ratio of 1:10. The Ecoflex solution consists of Ecoflex part A and part B with a mass ratio of 1:1. The PDMS solution consists of a 1:10 mass ratio of curing agents and PDMS monomers. The Ecoflex and PDMS solutions are mixed and degassed in a planetary mixer under vacuum pressure (DAC 600.2 VAC-P, Hauschild & Co. KG) (350 mbar for 1 min at 0 rpm, 20 s at 1500 rpm, and 20 s at 2350 rpm). Then the Ecoflex/PDMS solution is cured in an oven at 60 °C or 80 °C for 30 min. The blue elastomer films are fabricated using the above process, but 2 wt% additional coloring powder (Pigment powder, Vitarie) was added to the Ecoflex/PDMS solution via mixing.

### Shear modulus of the elastomer

The shear modulus of Ecoflex/PDMS composite is obtained by fitting the stress-strain data obtained from a uniaxial tensile test (Supplementary Fig. 36a, strain rate 40% min⁻¹). The specimen geometry is based on the European Standard EN ISO 527-2:1996 (type 5 A). Under the assumption of an incompressible Neo-Hookean hyperelastic model, the shear modulus of the Ecoflex/PDMS composite is $66.5 \pm 1.0$ kPa.

### Liquid metal and electrodes

The liquid metal (LM), also known as "Galinstan", consists of gallium, indium and tin with a mass ratio of 69: 22: 9 (Smart Elements, smart-elements GmbH). The mass density and electrical resistivity of Galinstan are about $6.44$ g/cm³ and $2.89 \times 10^{-7}$ Ω m, respectively, at room temperature. The electrodes inserted into the SEMA/SEMR (Fig. 1, Supplementary Fig. 4) are tin-coated copper wire (No. 0601025, Kabeltronik) with a diameter of 150 μm. Current is supplied to the robot via two 50-μm-thin copper wires (No. 1570224, TRU Components), in the case of the tethered robot.

### Magnets

Two magnets made of NdFeB (N45) are used in the experiments. Magnet 1: a circular plate magnet (SM-100×30-N, magnets4you GmbH) with a dimension of ∅ 100 × 30 mm. Magnet 2: two identical plate magnets (3965, EarthMag GmbH) stacked together with an overall dimension of ∅ 120 × 100 mm. Magnet 2 is only used in the experiments with SEMR UL and UR2. We obtained the magnetic fields of the two magnets from both experiments and simulations. Results are provided in the Supplementary Text.

### Robot feet

The sawtooth-shaped and rectangular feet of the robot are made by cutting 75 μm polyimide foils (300HN, Kapton) into the desired shape (Supplementary Fig. 13g) with a laser cutting machine or a scalpel. The L-shaped feet (Supplementary Fig. 24a, b, Supplementary Table 3) are 3D printed with an SLA printer (Form 3, FORMLABS). For the two-module SEMR, the feet are further modified to fit the rubber surface (Fig. 4f, Supplementary Fig. 28a, b).

### 3D printed substrate

The sawtooth substrate is printed by a commercial FDM printer (3 Extended, Ultimaker). The geometry of the substrate can be found in Fig. 1a and Supplementary Fig. 18. The substrate material is ABS Pro filament (No.1528301, Renkforce) and the printed layer thickness is 0.06 mm.

### Various substrates

Sandpapers: the grit sizes of the sandpapers are 80 (Sp-p80), 180 (Sp-p180) and 400 (Sp-p400) from kwb Germany GmbH; PDMS: the PDMS solution consists of a 1:10 mass ratio of curing agents and PDMS monomers, which is poured into a 3D printed mold and cured in the oven at 60 °C for 2 hours. The thickness of the PDMS film is 2 mm; Paper: office A4 paper. Wood: cut from a piece of balsa slat

(No. 1436844, Pichler); Metal: the surface of magnet 1; Plastic: cut from a polystyrene Petri dish (391-0556, VWR); Glass: regular glass plate with a thickness of 1.9 mm. Rubber: latex resistance bands (Silver, THERABAND)

## LM 3D printing system

The structure and main parts of the 3D direct ink writing (DIW) printer are shown in Supplementary Fig. 2, which mainly consists of a pressure dispenser (Ultimus V, Nordson EFD) and a commercial fused deposition modeling (FDM) printer (CR-10 V2, Creality). The dispenser is connected to a syringe (Optimeter optimum 30CC, Nordson EFD) mounted on the FDM printhead connected with a tapered tip featuring the inner diameter of 410 μm (7018298, Nordson EFD) or 200 μm (7018417, Nordson EFD). The 200-μm tip is only used for SEMR UL. A compressed air pipeline and a vacuum pump are connected to the dispenser to eject and hold the liquid metal, out of and in the syringe, respectively. The dispenser and FDM printer are controlled by a single board computer (4 Model B, Raspberry Pi) which runs a customized version of the application, OctoPrint. A small turbofan accelerates the oxidation of the surface of the liquid metal trace and a microscope tunes the gap of the tip to the printing surface. The gap between the nozzle and the printing surface is about 0.1 mm. The G-codes for printed patterns are generated manually or with customized script.

## Fabrication of SEMAs

In the fabrication of SEMAs, the LM is printed on an elastomer film with an initial thickness of around 350 μm made by spin-coating. The spin-coating parameters are provided in Supplementary Fig. 36b and Supplementary Table 6. A 400-μm-thick rubber frame is put around the LM as a mold for the uncured elastomer solution, which is poured on top of the printed LM traces. We use a glass slide to remove the excess elastomer solution. Subsequently, the bilayer film (with a layer of uncured elastomer solution) is degassed in a vacuum chamber (100 mbar) for several minutes until there are no air bubbles around the LM channels. Then the bilayer film is put in the oven (80 °C) for half an hour to cure the elastomer solution. After that, the metal electrodes are inserted into the bilayer film to connect the LM channels. Then a few droplets of elastomer solution are deposited around the location of the inserted electrodes to better seal the LM. The bilayer film is again put in the oven (60 °C) for half an hour to cure the few droplets of elastomer solution. Finally, the SEMA is cut from the bilayer film with a surgical blade. To obtain SEMA 2, a small square area (1.4 mm × 1.4 mm) is cut out from a SEMA 1 with the blade.

## Fabrication of bilayer films

The blue bilayer films are fabricated in six major steps (Supplementary Fig. 11). First, an elastomeric film is obtained by curing the mixed Ecoflex/PDMS/coloring powder solution (see section Materials and characterization) in a PMMA mold for 30 minutes at 60 °C. Then, a series of holes were cut out from the prepared elastomeric film with a laser cutter. This enables it to be mounted onto a 3D-printed frame for applying prestretches. Different frames were used for different target prestretches. To fabricate the top layer film, a 1 mm spacer was placed on the prestretched film and the mixed elastomer solution was poured inside. The excess solution above the spacer is removed from the frame by a sharp plastic blade. Another curing process of the top layer film runs for 30 minutes at 60 °C. Finally, a bilayer film is cut from the frame with a scalpel and becomes curved. Experimental results are illustrated in Supplementary Fig. 12.

## Fabrication of SEMRs

In the fabrication of SEMRs, the LM is printed on a stretched elastomer film fixed on a 3D printed frame (Fig. 1a, Supplementary Fig. 13). The following steps are similar to the fabrication of the SEMAs until the attachment of two feet with super glue (Ultra Gel Matic, Pattex). Details about these SEMRs are provided in Supplementary Text.

## PWM control system for actuation

The control system (Supplementary Fig. 6) consists of a Raspberry Pi 4 Model B single-board computer, an Adafruit 16-channel (PCA9685) PWM driver board, and multiple Digilent Pmod HB3 H-bridges. Both the PWM driver and the H-bridges are powered by the onboard 3.3 V regulator of the Raspberry Pi. The control software is written in Python and executed on the Raspberry Pi. Communication with the PWM driver is facilitated by an I2C bus. The PWM driver generates square-wave signals with specified frequency and duty cycle, as programmed. These output signals are used to drive the EN (enable) pins of the H-bridges. Additionally, Raspberry Pi GPIO pins are used to switch the direction (polarity) of the H-bridges. The H-bridge input terminals are connected to a benchtop power supply (GPO-33030, GW Instek), which makes it possible to limit the maximum current passing through the H-bridge. Each actuator is connected to one of the H-bridges' output terminals via thin magnet wires with a diameter of 0.05 mm (No. 1570224, TRU COMPONENTS).

## Bending and vibration test of the SEMAs

The SEMAs are placed vertically, perpendicularly to the surface of the magnet and clamped by a plastic support structure (Supplementary Fig. 7a). The lowest liquid metal channel of the SEMAs is about 6 mm away from the surface of the magnet. The SEMAs are actuated by the PWM control system. In the static bending test, a constant DC current is applied to the SEMA, while for the dynamic bending it is actuated by a square-wave current (Supplementary Fig. 7e).

## Numerical simulations

The numerical simulations are conducted with the commercial software ABAQUS/Standard (SIMULIA, Dassault Systèmes). To simulate the stretched bilayer film/SEMR, a user subroutine UMAT is used. Details about the subroutine are provided in the Supplementary Text, section 1.1. "Numerical simulation of the bilayer films". The neo-Hookean hyperelastic material model is used to model the mechanical behavior of the elastomer with a shear modulus of 66.5 kPa obtained from experimental characterization. The Poisson's ratio is set to be 0.49. In the simulation, the liquid metal is replaced by the elastomer for simplification, because of the low volume fraction of LM within the elastomer.

## Vibration test of the SEMRs

The robot is clamped by the copper wires to a supporting holder about 11 mm above the surface of the magnet (Supplementary Fig. 14a). The resonance frequency analysis was conducted with a function generator (33250 A, Agilent) as the signal source. The output of the function generator is fed to a custom-made buffer amplifier circuit powered by a benchtop power supply (EA-PS2316-050, EA Elektro-Automatik), capable of delivering up to 5 A current. The actuator is driven directly by the amplifier output. A one Ohm high-power resistor (HS150 1 R J, Arcol) is added in series to the actuator, serving as a shunt resistor. This allows measurement of the current waveform using a digital oscilloscope (GDS-11048, GW Instek) via Ohm's law. The function generator is controlled by a computer, and the frequency is incremented by 1 Hz every second. The response to both the sinusoidal and square-wave currents in the 1-100 Hz range has been measured this way. The vibration is recorded by a digital camera and the horizontal displacement of the robot feet is obtained by video analysis with a customized script (MATLAB, MathWorks).

## Speed measurement

The speed of the SEMRs is measured by analyzing the frames of the videos. To locate the robot in the frame, we first binarize the frame to

separate the body by tuning the threshold between white (body) and black (background). Then we get the position of the robot by calculating the center of the body area or the center of the boundary in the running direction. All these processes are done with a custom script (Python or MATLAB). The position of the robot can also be tracked by using the computer vision library OpenCV or software "Tracker" (https://physlets.org/tracker/). The measured speed is defined as the average speed over a period of 50 ms, unless stated otherwise.

## Robustness test
The experimental setup is similar to the walking and running test. The SEMR TL is put on the surface of the magnet. The robot is driven by the PWM control system with a square-wave current (0.2 A, 1 Hz). In Supplementary Movie 5, the SEMR TL recovers its operational capabilities, after being manually depressed and flattened by a plastic bar. The maximum force in this test is estimated by flattening the robot on a weighing scale (GP3202, Sartorius) and is equivalent to about 300 g.

The body of SEMR TL is also compressed by a tensile test machine (Z005, ZwickRoell) up to 2000 N. During the test, the resistance of the SEMR is recorded by a multimeter (2110, Keithley) using a 4-wire resistance measurement method.

## Thermal test
This setup is shown in Supplementary Fig. 8. A thermocouple is bonded to the center of the SEMA with super glue. The SEMA 1 is subjected to square-wave currents with amplitudes, 0.1 A, 0.3 A and 0.5 A. The SEMA is left to cool down to ambient temperature between the measurements. For each current, the measurement runs for 1000 s.

## Controller and power for the untethered robot
The small and medium PCBs (Supplementary Figs. 30 and 31) driving the untethered robot consists of a 555-variant timer integrated circuit (IC) (MIC1555, Microchip) and an n-channel metal-oxide-semiconductor field-effect transistor (MOSFET). The timer IC is configured as an astable multivibrator via an external resistor and a capacitor. This results in the generation of a square-wave signal at the output, with the resistor-capacitor (RC) time constant determining the frequency. The output pin is connected to the gate of the MOSFET and the contacts of the robot are connected to the drain. Power is provided by a Lithium-Polymer battery, which is specifically designed without ferromagnetic materials (no nickel foil). When the timer outputs a high signal, the MOSFET is switched on and the current flows through the robot.

The large PCB (Supplementary Fig. 32) addresses some of the downsides of the smaller versions, such as the lack of voltage regulation and the restriction to positive output voltages. This is facilitated by the addition of a motor driver IC with an integrated H-bridge (DRV8832, Texas Instruments). The square-wave output signal from the timer chip is fed into one of the two input pins of the driver IC. The other input pin is connected to the same signal through an inverter. The input pins, therefore, have opposite logic values at all times. This makes the driver IC change "direction" after every half-cycle of the input signal. The load is alternatingly connected to the battery voltage and the reverse battery voltage through the H-bridge circuitry. Voltage regulation is accomplished on the load side by switching to PWM modulation to keep the input voltage above a set value. This ensures that the driving frequency remains stable, while the output power is reduced. It also has the effect of preventing over-discharge of the battery.

We designed and drew all the PCBs with the KiCad EDA software (https://www.kicad.org/).

## Internal resistance of non-magnetic lithium battery
Three types of non-magnetic lithium batteries are used in this work (Supplementary Table 4). In the measurement, the battery is connected in series to a source meter (2611 A, Keithley) and in parallel to an oscilloscope (GDS-1104B, GW Instek). The source meter provides a 500 ms negative current pulse (device sinks current). The voltage is measured with the oscilloscope directly across the battery terminals (essentially 4-wire sensing). The first measurement is the open-circuit voltage $V_{open}$. The second measurement $V_{pulse}$ is taken 100 ms after the beginning of the pulse. The internal resistance of the battery is calculated by $R_{in} = (V_{open} - V_{pulse})/I$.

## Photography and video recording
Most optical microscopy images were recorded using a Nikon Eclipse LV100ND microscope. Supplementary Fig. 3g is recorded by the optical microscope BRESSER Erudit DLX (No. 5102000) with eyepiece camera BRESSER MikrOkular. Unless stated otherwise, a digital camera (EOS 80D, Canon) was used for photos and videos at a frame rate of 50 fps and a resolution of 1920 × 1080 pixels. The side view of the swimming robot SEMR TST was recorded by another camera (GC-PX10, JVC) at the same parameters. All the slow-motion videos were recorded by a high-speed camera (Chronos 2.1-HD, Krontech). Most of them were recorded at a frame rate of 2142 fps with a resolution of 1280 × 720 pixels. The experiment featuring the vibration of the SEMR subjected to two square-wave currents (0.5 A, 37 Hz) (Fig. 2h) was recorded at a frame rate of 4230 fps with a resolution of 1280 × 360 pixels. For the oscillation decay test (Supplementary Fig. 40), the video was recorded at a frame rate of 5406 fps with a resolution of 640 × 480 pixels.

## Data availability
All data needed to evaluate the conclusions in the paper are present in the paper and/ or the Supplementary Materials. Additional data related to this paper may be requested from the authors.

## Code availability
The code used in this paper is available upon any reasonable request.

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

## Acknowledgements
We thank Andreas Heiden, David Preninger, Reinhard Schwödiauer and Simona Bauer-Gogonea for the help in experiments and Christa Mit-schan for mental support. We also thank Yong Wang from Zhejiang University for the valuable discussion about vibrations. This work was supported by the ERC Starting Grant 'GEL-SYS' under grant agreement no. 757931 (M.K.).

## Author contributions
G.M., N.A. and M.K. conceived and initiated the project. G.M. designed the robots. D.S. built the electronic controllers. G.M. conducted the numerical analysis. N.A. and G.M. developed the theoretical framework. G.M., D.S., D.D., B.H., F.H., T.S. and M.D. conducted the experiments. G.M., D.S., D.D., B.H., N.A. and M.K. analyzed the results. G.M. and M.K. wrote the manuscript with comments and materials from all the authors. M.K. supervised the research.

## Competing interests
The authors declare no competing interests.
