## [Peer Review File · Nature Communications]

Ultrafast small-scale soft electromagnetic robotsEditorial Note: This manuscript has been previously reviewed at another journal that is not operating a transparent peer review scheme. This document only contains reviewer comments and rebuttal letters for versions considered at *Nature Communications* .

REVIEWERS' COMMENTS

Reviewer #1 (Remarks to the Author):

This manuscript has passed through several rounds of review and revisions. The current version is vastly improved over the original - I enthusiastically recommend publication.

Reviewer #2 (Remarks to the Author):

I acknowledge the improvements achieved through the new L-shaped feet in regard to being able to perform high-speed locomotion on variety of planar surfaces. The extent to which the authors were able to achieve high speed locomotion of a soft robot is very interesting. There were few major flaws with the previous design, that now have been addressed. Authors presented very sound experimental works and validations. I have no further comments on the technical parts anymore.

Author's Reply

We are very grateful to the reviewers for their invaluable and constructive comments. Responses to all the reviewer's comments are below.

Itemized response to the reviewers

Reviewer #1

This manuscript has passed through several rounds of review and revisions. The current version is vastly improved over the original - I enthusiastically recommend publication.

Answer:

We thank the reviewer for all the valuable suggestions which helped improve this work. We also thank for her/his appreciation of our revision.

Reviewer #2

I acknowledge the improvements achieved through the new L-shaped feet in regard to being able to perform high-speed locomotion on variety of planar surfaces. The extent to which the authors were able to achieve high speed locomotion of a soft robot is very interesting. There were few major flaws with the previous design, that now have been addressed. Authors presented very sound experimental works and validations. I have no further comments on the technical parts anymore.

Answer:

We thank the reviewer for all the valuable suggestions which helped improve this work. We also thank for her/his appreciation of our revision.